# Polymer Biocompositions and Nanobiocomposites Based on P3HB with Polyurethane and Montmorillonite

**DOI:** 10.3390/ijms242417405

**Published:** 2023-12-12

**Authors:** Beata Krzykowska, Anna Czerniecka-Kubicka, Anita Białkowska, Mohamed Bakar, Karol Hęclik, Lucjan Dobrowolski, Michał Longosz, Iwona Zarzyka

**Affiliations:** 1Department of Organic Chemistry, Faculty of Chemistry, Rzeszów University of Technology, Powstańców Warszawy 6, 35-959 Rzeszów, Poland; michal.longosz98@gmail.com; 2Department of Experimental and Clinical Pharmacology, Medical College of Rzeszow University, The University of Rzeszow, al. Tadeusza Rejtana 16C, 35-310 Rzeszow, Poland; aczerniecka@ur.edu.pl; 3Faculty of Chemical Engineering and Commodity Science, University of Technology and Humanities, Chrobrego 27, 26-600 Radom, Poland; a.bialkowska@uthrad.pl (A.B.); m.bakar@wp.pl (M.B.); 4Department of Biotechnology and Bioinformatic, Faculty of Chemistry, Rzeszów University of Technology, Powstańców Warszawy 6, 35-959 Rzeszów, Poland; kheclik@prz.edu.pl (K.H.); ldobrowolski@prz.edu.pl (L.D.)

**Keywords:** natural polyesters, modification, structure-properties relationship

## Abstract

Due to the growing interest in biopolymers, biosynthesizable and biodegradable polymers currently occupy a special place. Unfortunately, the properties of native biopolymers make them not good enough for use as substitutes for conventional polymers. Therefore, attempts are being made to modify their properties. In this work, in order to improve the properties of the poly(3-hydroxybutyrate) (P3HB) biopolymer, linear aliphatic polyurethane (PU) based on 1,4-butanediol (BD) and hexamethylene 1,6-diisocyanate (HDI) was used. The conducted studies on the effect of the amount of PU used (5, 10, 15 and 20 m/m%) showed an improvement in the thermal properties of the prepared polymer blends. As part of the tested mechanical properties of the new polymer blends, we noted the desired increase in the tensile strength, and the impact strength showed a decrease in hardness, in particular at the presence of 5 m/m% PU. Therefore, for further improvement, hybrid nanobiocomposites with 5 m/m% PU and organically modified montmorillonite (MMT) (Cloisite 30^®^B) were produced. The nanoadditive was used in a typical amount of 1–3 m/m%. It was found that the obtained nanobiocomposites containing the smallest amount of nanofillers, i.e., 1 m/m% Cloisite^®^30B, exhibited the best mechanical and thermal properties.

## 1. Introduction

Plastics are used in almost every field, from the automotive industry to medicine. With the growing awareness of environmental protection in society, biopolymers are increasingly attracting attention. These materials are replacing the previously widely used non-biodegradable polymers. Most research is based on already existing biomaterials such as polyhydroxyalkanoates (PHAs) [1,2]. These polymers can be converted into water and carbon dioxide in the presence of oxygen, and into methane in the case of anaerobic conditions.

PHAs, i.e., linear biopolyesters composed of hydroxyalkanoate units, are biodegradable and biocompatible, and can successfully replace petroleum-based materials. The most popular polymer in the PHAs family is poly(3-hydroxybutyrate), P3HB (Figure 1) [3], known as a double green polymer, which is used in the agricultural, packaging and medical industries [4,5,6,7].

Compared to other biodegradable polyesters, P3HB is a semicrystalline material with a high melting point (Tm = 173–180 °C) and its glass transition temperature (Tg) is about 0–10 °C. P3HB is produced as an energy carrier by various bacteria [2,8,9], which build polymer chains that are perfectly linear and isotactic [10], allowing unique properties and a high degree of crystallinity [11]. In this form, P3HB is biobased, biodegradable and biocompatible [12,13]. The polymer is UV-resistant, and is insoluble in water and relatively resistant to hydrolysis, which distinguishes it from other biodegradable polymers that are either water-soluble or moisture-sensitive.

Therefore, P3HB is a very interesting material, serving as a substitute for synthetic polypropylene. P3HB can be extruded, injected and pressed using conventional processing equipment. Unfortunately, the storage of P3HB products at room temperature causes deterioration of product properties, and the material becomes very brittle due to the formation of significant proportions of the crystalline phase [14]. Such disadvantages of P3HB, i.e., stiffness and brittleness, and above all low thermal stability, which is only slightly higher than its melting point [15], limit the large-scale commercial use of P3HB. The thermal instability of this polymer during plasticization renders the replacement of commercial non-biodegradable polymers with native P3HB difficult due to the narrow window of processing conditions.

In order to improve the properties of P3HB and increase its range of applications, the polymer is subject multiple modifications [16]. The manufacture of polymer blends and composites based on the P3HB matrix leads in most cases to the desired separation of its melting point and degradation temperature, which is characterized by better thermal properties and at the same time better mechanical properties.

The use of a polyurethane modifier will not only allow the desired modification of the thermal and mechanical properties of P3HB and its copolymers, but also accelerate its biodegradation, as the addition of hydrophilic polymers increases the absorption of water into the polymer mass and accelerates its hydrolysis [17]. In addition, polymer compositions with thermoplastic characteristics produced based on P3HB will still be highly biocompatible similar to native P3HB.

In order to improve the properties of P3HB, especially for thermal and mechanical applications, polymer compositions and nanocomposites based on it were produced. The study deals with the preparation of polymer compositions involving P3HB and aliphatic linear polyurethane (PU) obtained by reacting hexamethylene 1,6-diisocyanate (HDI) with 1,4-butanediol (BD) and hybrid nanobiocomposites based on the P3HB matrix with the abovementioned PU modifier with organically modified montmorillonite (Cloisite^®^30B), as well as studying the morphology, nanostructure, and thermal and mechanical properties of the obtained materials. Due to the better properties, new elasticized polymer compositions and composites can be used, e.g., in biodegradable packaging materials and gardening and construction materials.

## 2. Results

### 2.1. Polymer Blends of Poly(3-hydroxybutyrate)—Polyurethane

Polyurethane synthesized by reacting BD with HDI (Figure 2) was used to produce polymer compositions with P3HB. In order to improve the mechanical and thermal properties of aliphatic polyester, 5, 10, 15 and 20 m/m% of PU were used, as shown in Table 1.

The polymer compositions were prepared by melting homogenization using a co-rotating twin-screw extruder. A native P3HB was also processed to obtain a reference material under comparable conditions.

### 2.2. Spectral Analysis of the Produced Polymer Compositions

Figure 3 shows Fourier transform infrared (FTIR) spectra of P3HB, PU and their blends. The FTIR spectrum of P3HB illustrates the characteristic valence vibration band of the carbonyl group in the ester structure at 1718 cm^−1^ and the vibration bands of asymmetric and symmetric C–O bonds in the ester at 1271 and 1129 cm^−1^. Bands above 3000 cm^−1^ are not present.

On the other hand, the FTIR spectrum of PU (Figure 3) shows the valence vibration bands of N–H bonds of the urethane group above 3100 cm^−1^. The valence vibrations of the carbonyl group in the urethane group generate a band at 1704 cm^−1^, while the vibrations of the CO–NH bonds are visible at 1560 cm^−1^. Bands resulting from symmetric and asymmetric vibrations of the C–O bond in the urethane group are visible at 1247 and 1132 cm^−1^ (Figure 3).

In the FTIR spectra of P3HB–PU blends for wavenumbers above 3000 cm^−1^ there was a change in the shape of the band compared to PU’s spectrum. Its intensity decreased significantly and broadened, which is due to the formation of hydrogen interactions involving the urethane groups and/or terminal hydroxyl groups of aliphatic PU with the ester groups of P3HB (Figure 4) [18,19].

The intensity of the band above 3000 cm^−1^ increased with the PU content in the polymer mixture (Figure 3). In the range of 2800–3000 cm^−1^, there were three bands of asymmetric and symmetric C–H bonds of methyl and methylene groups at 2877, 2933 and 2977 cm^−1^, similar to the spectrum of P3HB or PU. Only one broadened valence vibration band of the carbonyl groups in the ester and urethane groups was observed at 1719 cm^−1^. Asymmetric and symmetric C–O bond vibration bands occurred together for the ester and urethane groups at 1268 cm^−1^ and 1129 cm^−1^.

### 2.3. Morphology Analysis of the Obtained Polymer Compositions

Scanning electron microscope (SEM) micrographs of the fractured surfaces at the point of load application were taken to investigate the changes in the morphology of the prepared polymer mixtures compared to the polymer P3HB matrix. The surface of the base P3HB, shown in Figure 5, had a slightly wavy, glassy morphology, which, along with several edge lines, indicates the presence of a regular crack propagation path. Such a morphology suggests a relatively brittle behavior.

Figure 6 shows SEM micrographs of the sampled fracture surfaces of the P3HB blends with 5, 10, 15 and 20 m/m% of a PU modifier. The presented images were obtained by scanning the fracture surfaces of the tested samples at the site of their cracking as a result of the applied load. The fracture surfaces of all P3HB–PU blends shown in Figure 6 are slightly wavy and glassy, indicating the presence of a regular and relatively linear crack propagation path, which is typical of brittle materials. As can be noted, the introduction of PU as a modifier caused an apparent disruption of the continuity of the P3HB matrix structure. One can also note the appearance of crystalline domains in the form of rough platelets arranged unidirectionally.

### 2.4. Mechanical Properties of the Prepared P3HB–PU Polymer Blends

Select mechanical properties of the obtained polymer blends were investigated by measuring, among others, the tensile strength and the relative elongation at break. Figure 7a shows the obtained strength results as functions of the amount of added PU modifier. The tensile strength of the P3HB–PU polymer blends depended on the PU modifier content, and it increased to a maximum value at 5 m/m% PU in the blend, after which it decreased with increasing PU content.

The values of relative elongation at break shown in Figure 7b have a similar tendency, i.e., they maximally increased after the addition of 5 m/m% PU, and then decreased up to a content of 15 m/m% PU. A further increase in the proportion of PU in the blend did not change the values of relative elongation at break.

As shown in Figure 7c, the impact strength (IS) of the obtained polymer blends also increased compared to unmodified P3HB, with a maximum value at 5 m/m% PU. However, a further increase in the PU content resulted in a decrease in the impact strength below that of native P3HB. The introduction of more than 15 m/m% PU no longer changed the impact strength of the blends. Similar findings were reported by Seydibeyoglu et al. [20] and by Jostet et al. [21]. Moreover, the hardness of the prepared P3HB was found to decrease with increasing modifier content (Figure 7d).

### 2.5. Thermal Stability of the Produced P3HB Polymer Blends

The thermal stability of P3HB polymer blends with 5, 10, 15 and 20 m/m% PU content was investigated by thermogravimetric analysis (TGA). The results of TGA are shown in Table 2, Figure 8. The polymer composition containing 5 m/m% PU started to degrade at 234 °C, which is 13 °C higher than the P3HB without a modifier. Adding more PU made the thermal stability even higher. The onsets of decomposition temperatures were 266, 268 and 268 °C for K10, K15 and K20, respectively.

The thermal degradation of the P3HB–PU polymer samples, just as with the unmodified P3HB, proceeded in a single step with the presence of a peak on the DTG curve representing the fastest decomposition at 278–280 °C. The mass of the residue at 600 °C did not exceed 1 m/m% (Table 2).

### 2.6. Thermal Properties of P3HB–PU Polymer Blends

In order to analyze the thermal properties of the prepared blends of P3HB and PU, DSC measurements were conducted. Figure 9 shows the dependence of the experimental heat flow of the tested blends as a function of temperature at a heating rate of 10 °C·min^−1^ in the temperature range of −90 °C to 195 °C, after prior cooling at the same rate in the temperature range given.

Qualitative thermal analysis was performed on the basis of heat flow rates of the semicrystalline P3HB and its blends with PU, recording only the glass transition (T_g_) and melting processes. Based on the analysis of glass transition region during heating, the change of specific heat (ΔCp) and the value of glass transition temperature T_g_ were determined. In turn, from the analysis of the melting region, the fusion heat ΔH_f_ and the melting temperature T_m (peak)_ were estimated. The occurrence of two peaks in the melting of polymers may indicate the coexistence of different crystalline forms or may be due to the introduction of PU. Two melting peaks were observed in thermograms of all polymer samples. The first, always smaller endothermic peak is due to the melting of less-stable crystals (T_m1 (peak)_, Table 3), while the second larger peak (T_m2 (peak)_, Table 3) corresponds to the melting of larger, well-formed crystals.

The estimated thermal parameters of the phase transitions are presented in Table 3. The crystallization heat ΔH_c_ and the crystallization temperature T_c_ were determined during the cooling analysis. The presence of PU caused a decrease in the crystallization temperature, which regularly decreased with an increase in the amount of PU introduced (Table 3).

### 2.7. Production of Hybrid Nanobiocomposites

Considering the obtained results of P3HB–PU polymer blends and with a view towards the further improvement of the physical properties of P3HB, hybrid nanobiocomposites were obtained by using polyester as a matrix, with the addition of the abovementioned aliphatic linear PU as a modifier and an organic-modified montmorillonite, Cloisite^®^30B, as a nanofiller. PU at 5 m/m% and Cloisite^®^30B at 1, 2 and 3 m/m% were used to produce the nanocomposites.

### 2.8. FTIR Analysis of Hybrid Nanobiocomposites

FTIR spectra of the hybrid nanobiocomposites based on the P3HB matrix with PU and Cloisite^®^30B as modifiers are shown in Figure 10. A small band above 3000 cm^−1^ appeared in the FTIR spectrum of the hybrid nanobiocomposites, whose intensity increased with increasing organic nanoclay content. Three bands of vibration of the asymmetric and symmetric C–H bonds of the methyl and methylene groups at 2877, 2933 and 2977 cm^−1^ and whose intensity increased with the increase in the amount of the nanofiller were observed in the range of 2800–3000 cm^−1^. One common valence vibration band of ester and urethane C=O groups was observed at 1707 cm^−1^ and at a lower frequency than in the case of the base P3HB matrix or the P3HB–PU blend (cf. Figure 3 and Figure 10), confirming the formation of intermolecular hydrogen bonds with Cloisite^®^30B. A common band for esters and urethanes for the vibration of asymmetric C–O bonds appeared at a wavenumber of 1270 cm^−1^, and a band for the vibration of symmetric C–O bonds appeared at 1097 cm^−1^, which was similar to the spectra of the P3HB–PU polymer blends.

### 2.9. Structure Analysis of Hybrid P3HB–PU–Cloisite^®^30B Nanobiocomposites

In order to characterize the nanoclay structure in the obtained biocomposites, SAXS measurements were conducted in the range from 1° to 28°, with the most diagnostic area being the range of 1–5°. Figure 11 shows the SAXS patterns of the prepared nanobiocomposites as well as the patterns of the unmodified P3HB and that of the Cloisite^®^30B nanoclay. The unmodified P3HB shows only background scattering at an angle 2Θ value smaller than 12° [22,23]. By contrast, the pattern of Cloisite^®^30B shows a peak with a maximum angle 2θ value of about 5.00° [24], which indicates a distance between the aluminosilicate plates in Cloisite^®^30B of approximately 1.77 nm. The tested hybrid nanobiocomposites with 1 m/m% organic nanoclay (the sample designated as K5-1) and higher amounts did not show a peak with a maximum angle 2θ value smaller than 5°. The absence of a peak in this area proves the complete delamination of Cloisite^®^30B, which means that the polyester and polyurethane chains not only filled the interlayer spaces of the aluminosilicate stacks, but led to their complete delamination [25].

### 2.10. Nanostructure Analysis of Produced Hybrid Nanobiocomposites

The nanocomposites were studied using the transmission electron microscope (TEM) method to observe the effects of nanoclay content on nanostructure features. Selected TEM micrographs of the nanobiocomposites are presented in Figure 12. The dark lines represent cross-sections of the nanoclay layers, and the gray area corresponds to the polymer matrix. Figure 12a shows the base polyester matrix, and Figure 12b–d show the structure of the hybrid nanobiocomposites containing 1–3 m/m% Cloisite^®^30B.

Analysis of the nanocomposite structure using TEM confirmed the conclusions of the SAXS analysis of the aforementioned composites. The TEM micrographs (Figure 12) show that Cloisite^®^30B is quite susceptible to deagglomeration and dispersion in the P3HB matrix, which is induced by the shear forces. As the nanofiller content decreases, the size of dispersed organic clay areas becomes smaller and smaller. The study of the image in Figure 12a reveals a homogeneous dispersion of clay platelets throughout the matrix on a nanometer scale. The MMT layers are well-dispersed in the matrix in the form of randomly stratified nanosheets, mainly indicating an exfoliation structure (blue arrows) [26]. Figure 12c,d show TEM images of the hybrid nanobiocomposites with 2 and 3 m/m% Cloisite^®^30B, respectively. A small share of nanosheet agglomeration is visible, and a partially intercalated structure (red arrows) is observed in Figure 12c.

### 2.11. Morphology Analysis of the Obtained Hybrid Nanobiocomposites

Figure 13 shows SEM micrographs of the fracture surfaces of the P3HB matrix nanocomposite samples containing 5 m/m% of the PU modifier and different amounts of the Cloisite^®^30B nanofiller, namely 1 m/m% (sample K5-1), 2 m/m% (sample K5-2) and 3 m/m% (sample K5-3). The incorporation of nanoparticles into the PU flexibilized P3HB (Figure 13) resulted in a noticeable roughness in the fracture surfaces of the nanobiocomposites. The polyester matrix modified with 5 m/m% PU and the smallest amount of 1 m/m% Cloisite^®^30B was characterized by a noteworthily completely different morphology from that of the other samples (Figure 13, K5-1). It is possible to observe distinct wavelike domains arranged in different directions, indicating the presence of a regular crack-propagation path. These domains are larger in size and more widely spaced than in the other samples. However, the addition of more than 1 m/m% of Cloisite^®^30B (i.e., 2 m/m% or 3 m/m%) already excludes the sample K5-1 from the formation of the mentioned agglomerates. This suggests that the addition of 2 m/m% or 3 m/m% Cloisite^®^30B is already pointless. The structure of samples containing 2 m/m% or 3 m/m% of Cloisite^®^30B (K5-2 and K5-3, respectively, Figure 13) became similar to that of a binary polymer mixture, i.e., containing P3HB and a polymeric modifier (Figure 6).

### 2.12. Mechanical Properties of the Obtained Nanobiocomposites Hybrids

As a part of the mechanical tests, the tensile strength, relative elongation at break, impact strength and hardness of the obtained hybrid nanobiocomposites were investigated similarly to the P3HB–PU polymer blends, and the results are shown in Figure 14. The effect of nanofiller addition on the increase in tensile strength is shown in Figure 14a. As mentioned earlier (Figure 7a), the addition of PU resulted in a slight increase in tensile strength, while the incorporation of organic nanoclay caused a noticeable increase in tensile strength (TS). Increasing the amount of nanoadditive caused a decrease in TS, which intensified with an increase in the nanofiller content [27]. It should be noted that the tensile strength of all hybrid nanobiocomposites was higher than that of the P3HB matrix. Nanoclay (Cloisite 30B) in an amount between 1 and 3 m/m% had good dispersibility in the polymer matrix of PH3B-PU, which improved the mechanical properties of the biocomposites [28,29,30,31]. The simultaneous addition of 5 m/m% PU and 1 m/m% Cloisite^®^30B (K5-1) resulted in a large increase in relative elongation at break of about 58% compared to the P3HB matrix sample (Figure 14b). A further increase in the percentage of modified nanoclay to 3 m/m% led to a decrease in the value of elongation at break, even below the value for the unmodified P3HB (K5-3).

As in the case of the impact strength of the hybrid nanobiocomposites (Figure 14c), we observed that the simultaneous addition of PU and Cloisite^®^30B (1 m/m% of nanoclay) led to a significant increase in impact strength, by approximately 30% compared to the unmodified P3HB matrix. However, the impact strength decreased for the nanobiocomposite samples designated K5-2 and K5-3, with values still higher than that of the reference sample with 3 m/m% of nanoclay.

A decrease in the hardness of the tested nanobiocomposites was observed (Figure 14d). The addition of polymeric modifier made the polyester/polyurethane blend more flexible, and the greatest decrease in hardness of about 11% was observed for the sample K5 (i.e., the P3HB containing 5 m/m% PU).

### 2.13. Thermogravimetric Analysis of Nanobiocomposites

The prepared hybrid nanobiocomposites were subjected to thermogravimetric analysis in order to investigate the physicochemical changes that occurred during heating. The results of the thermogravimetric analysis of the nanobiocomposites are shown in Table 2 and Figure 8. The addition of 1 and 2 m/m% nanofiller resulted in a 24 °C increase of the degradation temperature. By contrast, the addition of 3 m/m% nanoclay (K5-3) resulted in a slight (4 °C) decrease in the thermal stability of the hybrid nanobiocomposite. However, the thermal stability of this nanocomposite was still higher than that of the unfilled P3HB and the polyester modified with only 5 m/m% PU (K5). The temperature of the maximum decomposition rate was similar for the unmodified P3HB and its nanobiocomposites and oscillated between 279 and 276 °C.

### 2.14. Thermal Analysis Based on the Differential Scanning Calorimetry Measurements

Figure 15a shows the heat flow rates versus temperature in the range of −90 °C to 195 °C for the K5-1, K5-2 and K5-3 nanobiocomposites obtained based on the DSC measurements.

The glass transitions were observed in the range of 4.40–5.25 °C and the onset melting temperature was in the range of 134.6–156.7 °C. The changes in heat capacity and heat of fusion were also determined for the future estimation of the phase contents of new materials. Based on Table 3 and Table 4, the lowest value of the glass transition temperature was observed for the K5-2 nanobiocomposite, which allows us to assume that this material will be the most plasticized in reference to the series of materials considered. The greatest processing capabilities are offered by K5-3, because the processing window was estimated as 122.2 °C. For the remaining nanobiocomposites K5-1 and K5-2, it was 116.8 °C and 118.2 °C, respectively. Figure 15b shows a comparison of the heat flow of the poly(3-hydroxybutyrate) (P3HB), nanobiocomposite (K5-1) and polyurethane (PU) versus temperature. The PU heating scan presented the glass transition at T_g_ = 14.90 °C and ΔC_p_ = 0.3094 J·g^−1^·°C^−1^, the melting region at Tm onset = 166.30 °C, and the heat of fusion at ΔH_f_ = 90.39 J·g. It can be observed that DSC heating scans present a single glass transition and melting region, which indicates a complete miscible area of P3HB and PU in all nanocomposites.

## 3. Discussion

In order to improve the properties of P3HB, polymer compositions were produced with the use of linear aliphatic PU based on BD and HDI with the following amounts of PU: 5, 10, 15 and 20 m/m%.

FTIR was used to study the interactions between P3HB and PU. The FTIR spectra reveled a formation of hydrogen bonds between the urethane groups and/or terminal hydroxyl groups of the aliphatic PU with the ester groups of P3HB (Figure 4). Thus, the presence of hydrogen bonds decreases specific P3HB chain interactions (polyester–polyester interactions).

The SEM images of fractured samples allow to explain the mechanism of improvement of the mechanical properties of the nanobiocomposites tested by the added polymeric modifier. The observed morphology of compositions suggests the interaction of the biopolymer matrix with the polymeric modifier (PU), resulting in disentanglement of interacting P3HB chains [31]. These disentanglements may be related to an increase in the elongation and impact strength of the tested blends as the polymer modifier content increased, and the enhancement of the mentioned mechanical properties is due to the flexibilization of the polyester matrix by the PU modifier. If the phase separation is not detectable, the blends give optimum mechanical and thermal property results. Urethanes groups of PU and terminal hydroxyl groups can interact with P3HB ester groups, forming hydrogen bonds. The good ductile properties are achieved by the strong interactions exerted between the P3HB and PU chains [19,32].

The tested selected mechanical properties of the polymer compositions show that the composition containing the smallest amount of PU (K5) had the best properties, and the effect of worsening of the properties with increasing PU content was observed. Some authors have reported a trend of decreasing the tensile strength in several plasticized polymers with increasing plasticizer content with a slight increase in elongation at break [20,33,34]. This finding might be explained by the increase of the free volume in the system due to the addition of flexible chains of modifier material, leading to a reduction in specific polyester–polyester interactions. The maximum improvement of all measured mechanical properties with an addition of a small amount of polymeric modifier (i.e., 5 m/m% PU) can be attributed to the formation of a grafted interpenetrating polymer network structure, as reported elsewhere with similar crosslinkable systems. The exception means a decrease in hardness, which is desirable. This may be due to the softening effect and flexibilization of the samples induced by the consequent increase in the free volume in the mixture and the presence of flexible modifier chains [35].

The introduction of PU into P3HB caused an increase in the degradation temperature of the resulting polymer mixture compared to the unmodified P3HB. There was an increase in the degradation temperature of the obtained blends of 13–57 °C in relation to the native P3HB. The addition of plasticizer led to a noticeable improvement in the thermal stability of the P3HB, which was achieved via the interaction of the plasticizer molecules and the polyester chain. It resulted in the formation of a thin physical barrier on the surface of the blends and obstructed permeability of volatile products towards the exterior. It effectively retarded the thermal degradation of blends, as explained in other studies [36,37,38,39,40].

DSC analysis indicated that the melting onset temperature T_m (onset)_ of the polymer samples was lower than that of the unmodified P3HB and decreased regularly with increasing PU content in the mixture [35]. The difference between the melting point and the degradation point was above 100 °C for most of the obtained blends. A decrease in the glass transition temperature of the new polymer mixtures was also observed, which proves the plasticization of the polyester due to the introduction of PU. A similar effect was observed by Fernández-Ronco et al. [41] in P3HB—poly(butylene succinate-butylene dilinoleate) blends—and by Kozlowska et al. [42].

In order to further improve P3HB’s properties, hybrid nanobiocomposites were produced with the use of 5 m/m% aliphatic linear PU as a modifier and organic-modified MMT—Cloisite^®^30B—as a nanofiller at 1, 2 and 3 m/m%. FTIR analysis of the produced nanobiocomposites confirmed the formation of hydrogen bonds between the urethane groups of the modifier and the ester groups of the polyester and the modified MMT, as well as the compatibility of the components [43,44].

Analysis of SAXS patterns of the produced hybrid nanobiocomposites indicated that the exfoliated structure of the nanocomposites was obtained, guaranteeing a definite improvement in the properties of the new materials over those of the initial matrix. TEM analysis confirmed that the exfoliated layers are the main structure of the produced nanobiocomposites, as evidenced by single clay nanosheets and the absence of tactoids [45].

SEM analysis of the nanobiocomposites’ morphology suggests that the P3HB chains are interacting less with each other, with the effect of the P3HB becoming more flexible [46]. This may be related to the formation of P3HB–PU–MMT adducts that are easily displaced with respect to each other, and this phenomenon may affect the significant increase in elongation and the slight increase in impact strength of the analyzed sample. On the other hand, an increase in the hardness of the material may suggest a stiffening effect of MMT, which translates into the roughness of the observed nanocomposites. Increasing the amount of added nanoparticles above 1 m/m% does not affect the flexibility of material, which affects the decrease in impact strength, toughness and elongation at break of these nanobiocomposites and suggests the stiffening of P3HB–PU polymer blends by such amounts of Cloisite^®^30B (2 m/m% and 3 m/m%) [47].

The improvement in impact strength can be induced by the presence of P3HB–PU–MMT adducts identified in the SEM images, while the decrease in impact strength is related to the stiffening effect due to Cloisite^®^30B [48]. The introduction of nanofiller resulted in a relative stiffening of the plasticized structure and a slight increase in the hardness of nanocomposite slightly above the hardness value of the K5 sample. In the case of hybrid nanobiocomposites, it was noted that the lowest content of nanofiller (1 m/m%) added to 5 m/m% PU (K5-1) allowed the formation of P3HB–PU–MMT adducts, showing the best mechanical properties.

Thermogravimetric analysis of the nanobiocomposites (Table 2, Figure 8) showed that the addition of 5 m/m% aliphatic PU and 1 or 2 m/m% organic nanoclay resulted in a 24 °C increase in the stability of the nanobiocomposite compared to the reference P3HB sample [41,49]. Moreover, based on the difference between the degradation temperature and the onset melting temperature (the start of melting), the processing window of the nanobiocomposites was determined to be in the range of 116–120 °C.

## 4. Materials and Methods

### 4.1. Materials

P3HB was supplied by Biomer (Krailling, Germany); its weight-average molecular weight was M_w_ = 443,900 g·mol^−1^ and its dispersity index was (M_w_·M_n_^−1^) = 5.72; the P3HB melt flow index was 0.11 g·(10 min)^−1^ (180 °C at 2.16 kg). Organically modified montmorillonite (Cloisite^®^30B) was supplied by Southern Clay Products Inc. (Gonzales, LA, USA). Cloisite^®^30B is a natural montmorillonite modified with methylbis(2-hydroxyethyl)tallowalkylammonium cations. Hexamethylene 1,6-diisocyanate and dibutyltin dilaurate (DBTL) were purchased from Aldrich (Darmstadt, Germany), 1,4-butanediol, 98%, was purchased from Aldrich (Darmstadt, Germany) and acetone was purchased from Chemsolute (Bremgarten, Germany).

#### 4.1.1. Synthesis of Linear Polyurethane

First, BD dried acetone and DBTL were introduced into a three-necked round-bottom flask equipped with a mechanical stirrer, thermometer and dropper. Then, an appropriate amount of HDI was dropped into the mixture with a molar ratio of isocyanate groups to hydroxyl groups that should ideally be 1:1.08. The rate of dropping was adjusted to keep the temperature of the reaction mixture below 25 °C. The synthesis was monitored on the basis of isocyanate number determination according to the standard [50]. During the reaction (Figure 2), a product precipitated out of solution. The polyurethane was separated by filtration and was brought to a constant mass by exposure in a vacuum dryer at the temperature range of 40–100 °C. The molar weight of the PU was 109,000 g·mol^−1^ and the hydroxyl number was 277 mg KOH·g^−1^.

#### 4.1.2. Preparation of Polymer Blends

In order to prepare P3HB–polymer blends with PU, P3HB was dosed into a Stephan-type mixer, followed by PU in appropriate amounts, i.e., 5, 10, 15 and 20 m/m%. The mixture was homogenized by stirring at room temperature for about 20 min. Table 1 shows the composition of the resulting mixtures. The homogeneous mixture was then dosed into a hopper of a co-rotating twin-screw extruder with a screw operating diameter of D = 25 mm and L/D = 33, operating at a speed of 300–450 rpm. The different zones of the extruder were maintained during extrusion at the following temperatures: hopper—31–32 °C, II zone—124–134 °C, III zone—142–167 °C, IV zone—135–136 °C, V zone—135–136 °C, VI zone—148–149 °C, VII zone—148 °C, VIII zone—148-150 °C and IX zone—148–151 °C, and the head was kept at temperature range between 172 °C and 179 °C. During extrusion, volatile particles were discharged by atmospheric degassing, and the temperature of the extruder head and the heating zones of the extruder plasticizing system were kept constant. The melted composition was cooled in a cooling bath, pelletized and dried at 60 °C for 2 h.

#### 4.1.3. Preparation of Polymer Nanobiocompositions

Polymer nanobiocomposites based on a P3HB matrix with P3HB, 5 m/m% PU and 1, 2 and 3 m/m% organic nanoclay Cloisite^®^30B were prepared using a mixer, in which P3HB, PU and Cloisite^®^30B were mixed according to the composition given in Table 1. The mixing process lasted for about 20 min. The homogenized mixture was extruded using a co-rotating twin-screw extruder, under conditions similar to those described above in Section 4.1.2.

### 4.2. Analytical Methods

The isocyanate group contents were determined according to the standard [50]. The hydroxyl number was determined according to the standard [51].

#### 4.2.1. Small-Angle X-ray Scattering

The small-angle X-ray scattering technique was used to characterize the nanoclay structure in the prepared nanobiocomposites. The measurements were conducted at laboratory temperature using a Bruker SAXS Nanostar-U X-ray diffractometer (Dekendorf, Germany). The spectra of the samples were studied in transmission mode. The small-angle diffractometer was connected to a CuKa-filtered radiation source (1.54 Å) placed in a sealed tube, operating at 50 kV and 30 mA. A 2D detector (Vantec2000, Brucker, Dekendorf, Germany) was used to scan the entire surface of the sample using a spot beam of about 500 μm. The scanning range was determined by varying the distance of the sample from the detector. The detector’s resolution and angular range allowed measurements of 2048 × 2048 pixels. The measurements were conducted from 1 to 28° for a period of 2 h.

#### 4.2.2. Transmission Electron Microscopy

Transmission electron microscopy was used to study the nanostructure of the obtained biocomposites. The measurements were conducted on a TECNAI G12 Spirit-Twin instrument (Hillsboro, OH, USA) (LaB6 source) equipped with a FEI Eagle 4k CCD camera (Eindhoven, The Netherlands) operating at an acceleration voltage of 120 kV. Prior to analysis, the samples were cut with a cryoultramicrotome and placed on 300 mesh copper grids.

#### 4.2.3. Scanning Electron Microscopy

The extruded base P3HB, P3HB–PU polymer blends and prepared nanobiocomposites were studied using a JEOL-type JSM-6490 LV scanning electron microscope (Tachikawa, Tokyo, Japan) to analyze the morphology of the materials in the micro-area. First, the process of freezing the samples in liquid nitrogen was carried out, followed by their breakage using a blade. The samples thus prepared were coated with a layer of gold about 10 nm thick using a JEOL JFC-1300 gold sputtering machine (Tachikawa, Tokyo, Japan). Microphotographs showing the morphology were taken along with the surface structure of the P3HB matrix blends and nanobiocomposites.

#### 4.2.4. FTIR Spectroscopy

Infrared spectra of the base P3HB, PU, polymer blends and hybrid nanobiocomposites were measured using an ALPHA FTIR spectrometer (Brucker, Dekendorf, Germany) performing measurements at the wave numbers of 400–4000 cm^−1^. The spectra were recorded at a resolution of 0.01 cm^−1^ using the ATR technique.

#### 4.2.5. Mechanical Properties

The specimens for mechanical testing were obtained by injection molding using an Arburg 420 M injection molding machine of the Allrounder 1000-250 type (Dortmund, Germany). The process was carried out at 140–172 °C. The injection molding temperature was 25 °C for P3HB and 30 °C for polymer blends and nanobiocomposites.

The tensile mechanical properties were determined in accordance with [52] using an “Instron 4505” testing machine. Tensile strength and relative elongation at break were measured using a rate of 5 mm/min. Unnotched Charpy impact tests were performed in accordance with [53] using a Zwick 5102 impact hammer. Shore hardness was determined in accordance with [54] using a Zwick apparatus.

#### 4.2.6. Thermogravimetric Analysis

Thermogravimetric analysis (TGA) of P3HB and its blends as well as hybrid nanobiocomposites was carried out using a Mettler Toledo TGA/DSC 3+ thermogravimetric analyzer (Greifensee, Switzerland). The heating rate of samples was 5 °C·min^−1^ in the temperature range from +25 to +600 °C. The measurements were conducted in a nitrogen atmosphere. The following temperatures were determined: temperature of the beginning of decomposition (T_on_), temperature of half mass loss (50%), temperature of the maximum decomposition rate (T_max_) and the total mass loss of the sample at 600 °C.

#### 4.2.7. DSC Analysis

Differential scanning calorimetry measurements were conducted for PU, P3HB, its blends and nanobiocomposites using a differential scanning calorimeter, which provided results in the form of heat flux versus temperature as a response to a linear change in temperature over time. Measurements of the heat flow rate in the temperature range from −90 °C to 195 °C (from 183.15 K to 468.15 K) were carried out using a differential scanning calorimeter (DSC), a type of the Discovery DSC 2500, from TA Instruments, Inc. (New Castle, DE, USA). In each case, the analyses were carried out in an atmosphere of nitrogen with a constant flow rate of about 50 mL·min^−1^. Calibration of temperature and heat flux in the calorimeters was carried out with respect to the melting parameters of indium, i.e., the initial melting temperature, the so-called “onset,” of T_m (onset)_ = 156.6 °C (429.6 K), and the enthalpy of fusion HΔ_f_ = 28.45 J·g^−1^ (3.28 kJ·mol^−1^).

## 5. Conclusions

New polymer blends were prepared with P3HB and aliphatic linear PU. Linear PU synthesized with HDI and BD was used as modifier for P3HB in the amounts of 5, 10, 15 and 20 m/m%. The thermal stability of the P3HB–PU polymer compositions was higher than that of the unmodified P3HB matrix. The difference between the degradation temperature of P3HB and its blends with PU was 13–57 °C. The difference between the melting point and degradation temperature was above 100 °C for most of the polymer blends obtained. DSC analysis also showed a decrease in the glass transition temperature of the tested polymer blends. The mechanical properties of the new P3HB–PU polymer blends at 5 m/m% modifier content showed a desirable increase in tensile strength and relative elongation at break and impact strength, and a decrease in hardness. A further increase in the PU percentage in the composition resulted in a deterioration of mechanical properties except for the desired decrease in hardness.

P3HB–PU polymer blends containing 5 m/m% aliphatic linear PU showed the best mechanical and thermal properties, and were therefore used with different amounts of organically modified nanoclay (Cloisite^®^30B) to prepare hybrid nanocomposites based on P3HB.

Using the SAXS technique and transmission electron microscopy, it was found that polyester and polyurethane chains penetrated the interlayer spaces of the organically modified MMT (Cloisite^®^30B) and complete delamination occurred during direct mixing in the extrusion process. Spectral analysis confirmed the interaction between the polymers and the nanofiller and their compatibility in the obtained nanobiocomposites. The morphology of the prepared hybrid nanobiocomposites demonstrated the interaction of the biopolymer matrix with PU and Cloisite 30B as well as the formation of P3HB–PU–MMT adducts easily migrating in relation to each other and leading to improved mechanical properties compared to those of the base P3HB and P3HB–PU polymer blends.

The incorporation of nanoplatelets into the polymer matrix ensured a better thermal stability of the hybrid nanobiocomposites. There was an increase in the onset temperature of degradation by 24 °C relative to P3HB and 11 °C relative to the P3HB–PU polymer blends.

The hybrid nanobiocomposites based on the P3HB matrix have shown higher impact and tensile strength as well as higher relative elongation at break. The best mechanical properties were obtained by a nanobiocomposite containing 1 m/m% Cloisite^®^30B (a 30% increase in impact strength, an 11% increase in tensile strength and a 58% increase in relative elongation at break). The obtained hybrid nanobiocomposites containing the smallest amount of nanofillers—1 m/m% Cloisite^®^30B and 5 m/m% PU—were characterized by the best mechanical and thermal properties.

Due to the better thermal properties, i.e., a higher degradation temperature and a lower melting point compared to native P3HB, as well as a broader processing window, and better mechanical properties, especially impact strength, the hybrid nanobiocomposites based on P3HB and containing the smallest amount of nanoadditive are expected to be used in the production of biodegradable packaging materials. Their biodegradability will be the subject of a future paper.

## Figures and Tables

**Figure 1 ijms-24-17405-f001:**
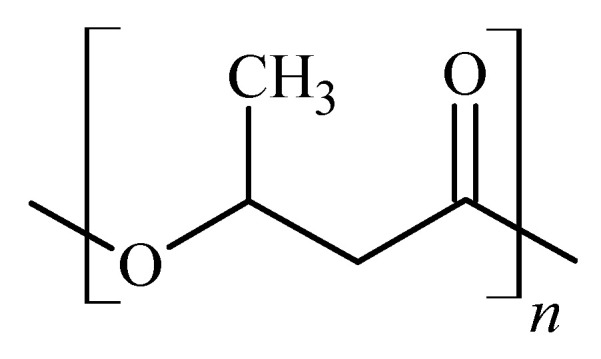
Repeated unit of P3HB.

**Figure 2 ijms-24-17405-f002:**
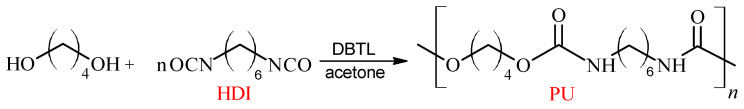
Synthesis scheme of polyurethane (PU).

**Figure 3 ijms-24-17405-f003:**
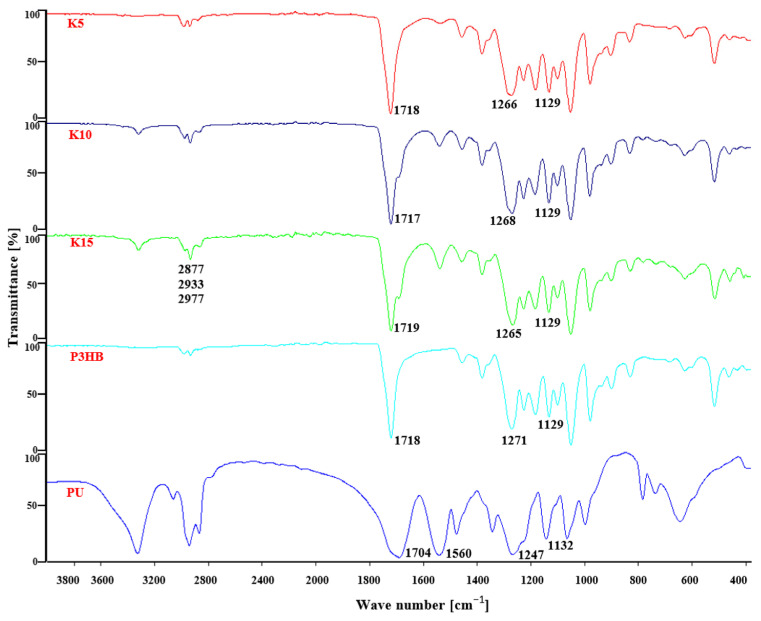
FTIR spectra of P3HB, PU and their blends containing 5, 10 and 15 m/m% PU.

**Figure 4 ijms-24-17405-f004:**
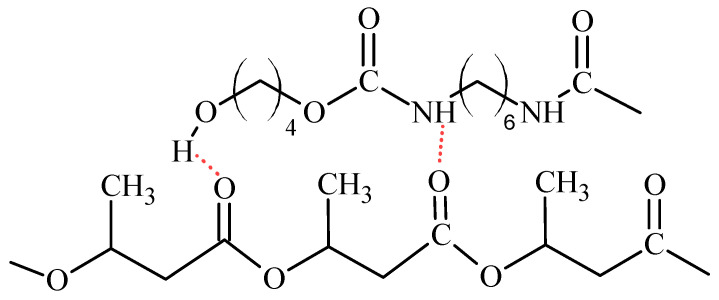
Scheme of hydrogen bond formation between P3HB and PU chains.

**Figure 5 ijms-24-17405-f005:**
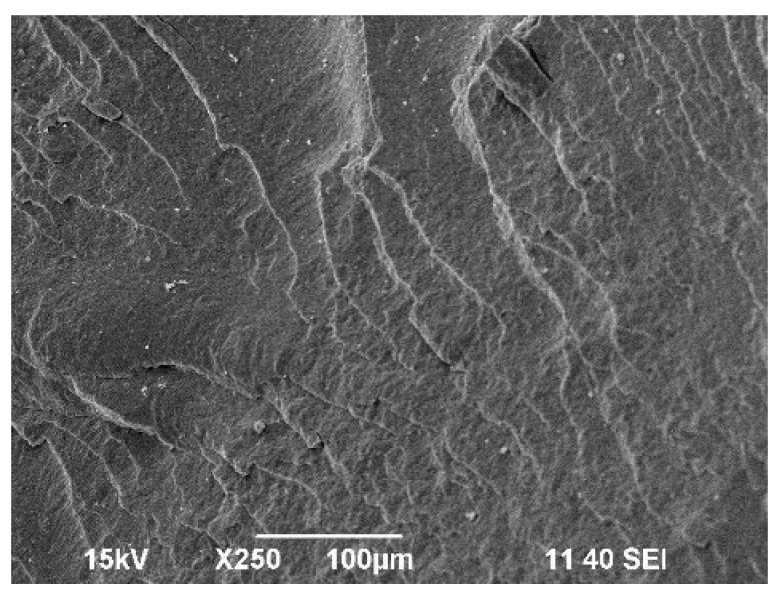
SEM micrograph of native P3HB.

**Figure 6 ijms-24-17405-f006:**
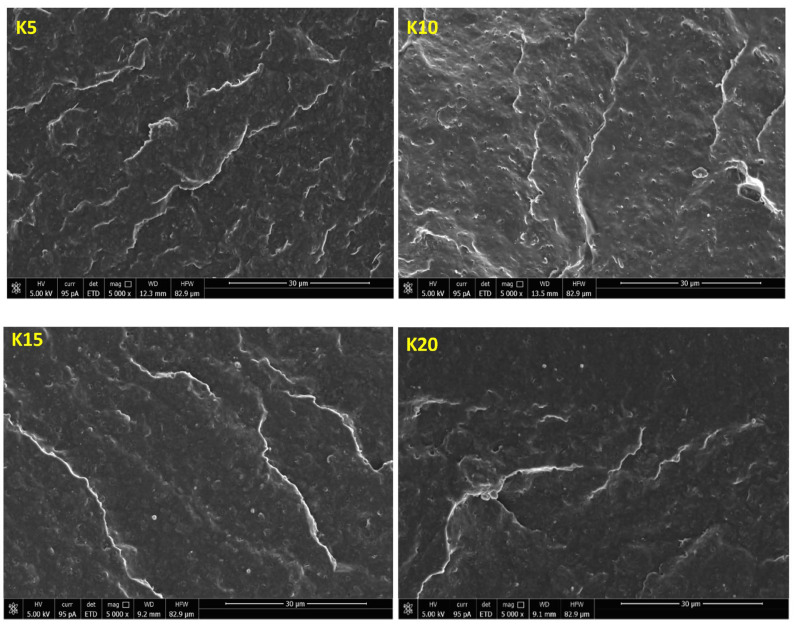
SEM micrographs of the P3HB polymer compositions containing 5, 10, 15 and 20 m/m% PU, designated as K5, K10, K15 and K20, respectively.

**Figure 7 ijms-24-17405-f007:**
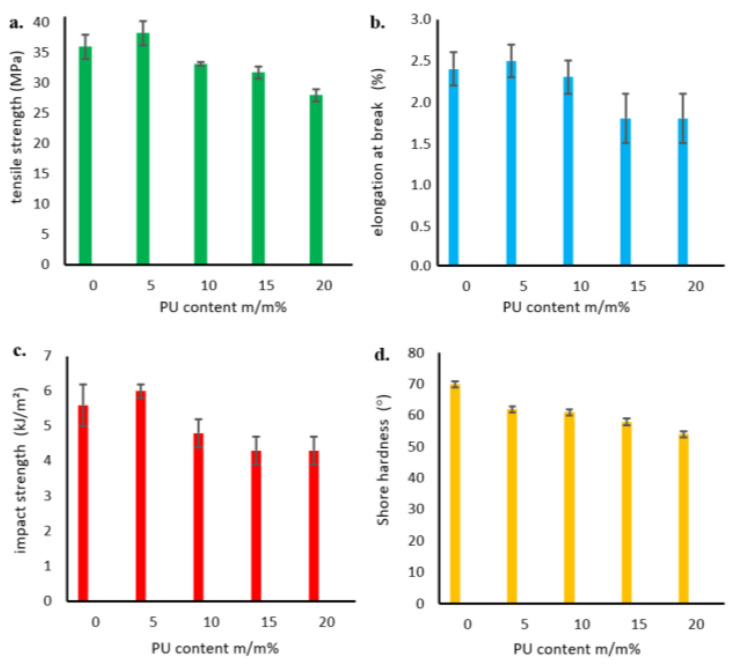
Tensile strength (**a**), relative elongation at break (**b**), impact strength (**c**), and hardness (**d**) of P3HB–PU polymer compositions as functions of the amount of PU.

**Figure 8 ijms-24-17405-f008:**
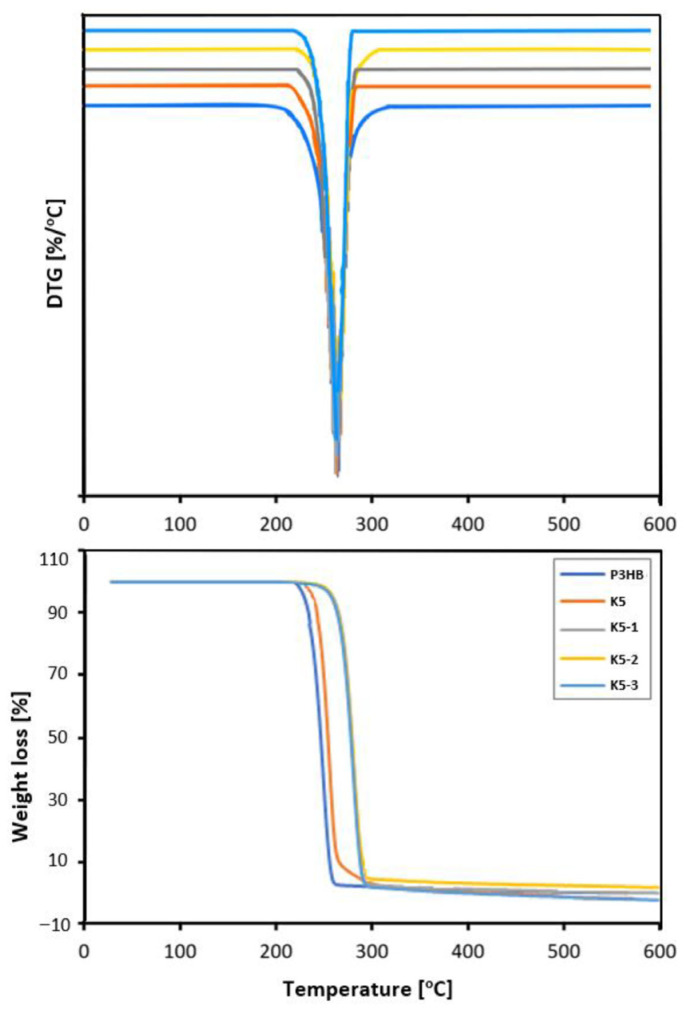
TG and DTG curves of the native P3HB, its composition with 5 m/m% of PU (K1) and composites of P3HB with 5 m/m% of PU and 1, 2 or 3 m/m% Cloisite 30B (K5-1, K5-2 and K5-3, respectively).

**Figure 9 ijms-24-17405-f009:**
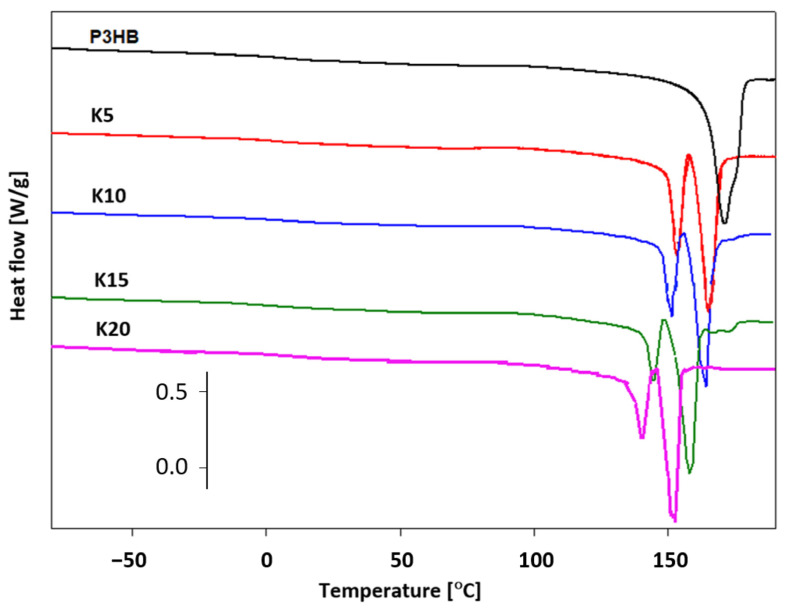
Comparison of heat flows of P3HB and its polymer compositions with PU as a function of temperature upon heating at a rate of 10 °C·min^−1^.

**Figure 10 ijms-24-17405-f010:**
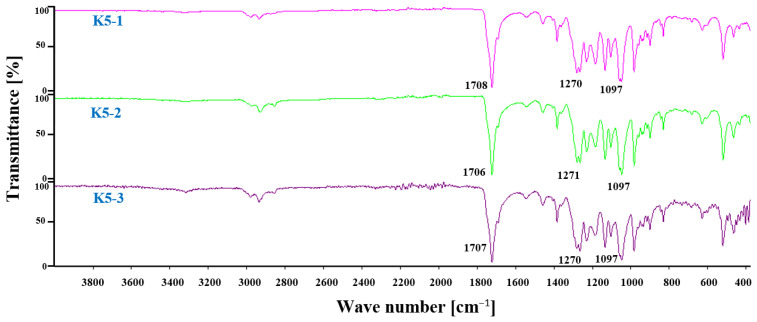
FTIR spectra of hybrid nanobiocomposites based on P3HB containing 5 m/m% PU and 1, 2 or 3 m/m% Cloisite^®^30B (K5-1, K5-2 and K5-3, respectively).

**Figure 11 ijms-24-17405-f011:**
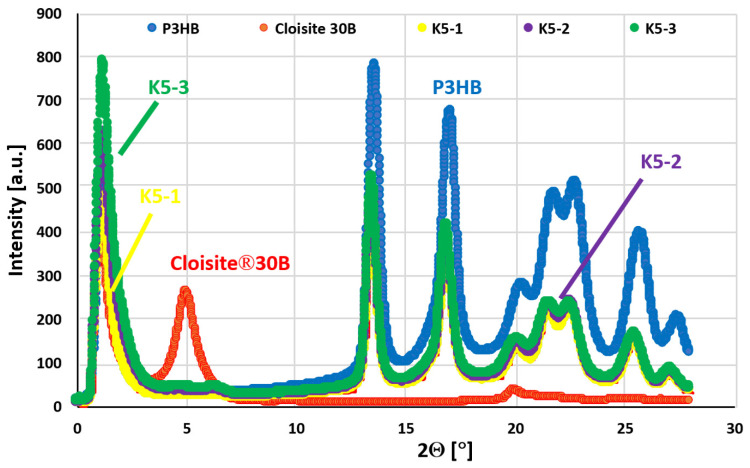
SAXS plots of native P3HB, Cloisite^®^30B (an organic nanoclay) and the produced hybrid biocomposites containing 5 m/m% PU and 1, 2, or 3 m/m% Cloisite^®^30B designated as K5-1, K5-2 and K5-3, respectively.

**Figure 12 ijms-24-17405-f012:**
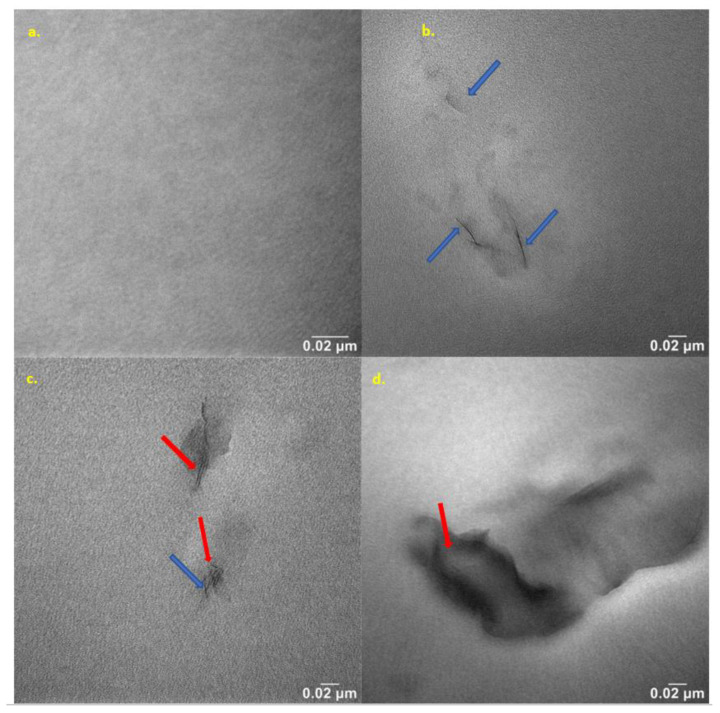
TEM microphotographs of P3HB (**a**) and its nanobiocomposites containing 5 m/m% PU and: (**b**) 1, (**c**) 2 or (**d**) 3 m/m% Cloisite^®^30B, designated respectively as K5-1, K5-2 and K5-3 (the visible artifacts resulted from the sample preparation).

**Figure 13 ijms-24-17405-f013:**
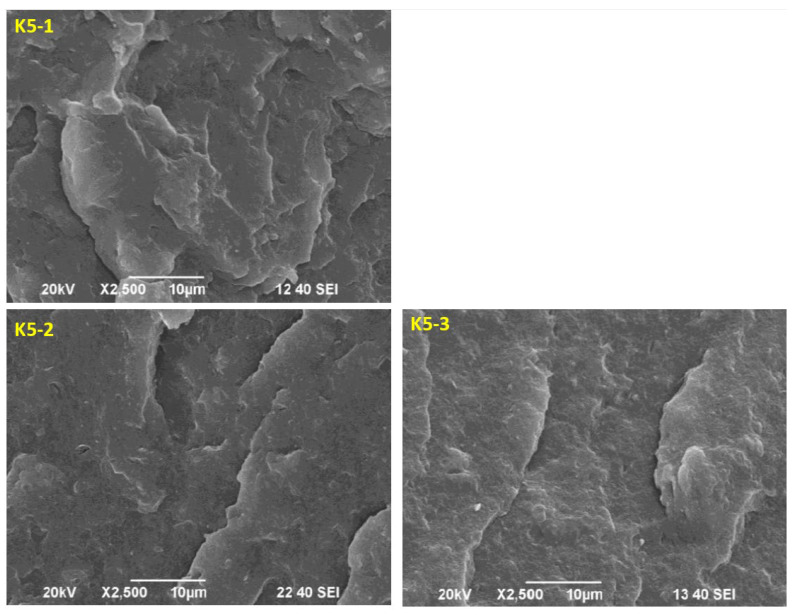
SEM micrographs of P3HB hybrid nanobiocomposites containing 5 m/m% PU and 1, 2 or 3 m/m% organic modified nanoclay—Cloisite^®^30B, designated K5-1, K5-2 and K5-3, respectively.

**Figure 14 ijms-24-17405-f014:**
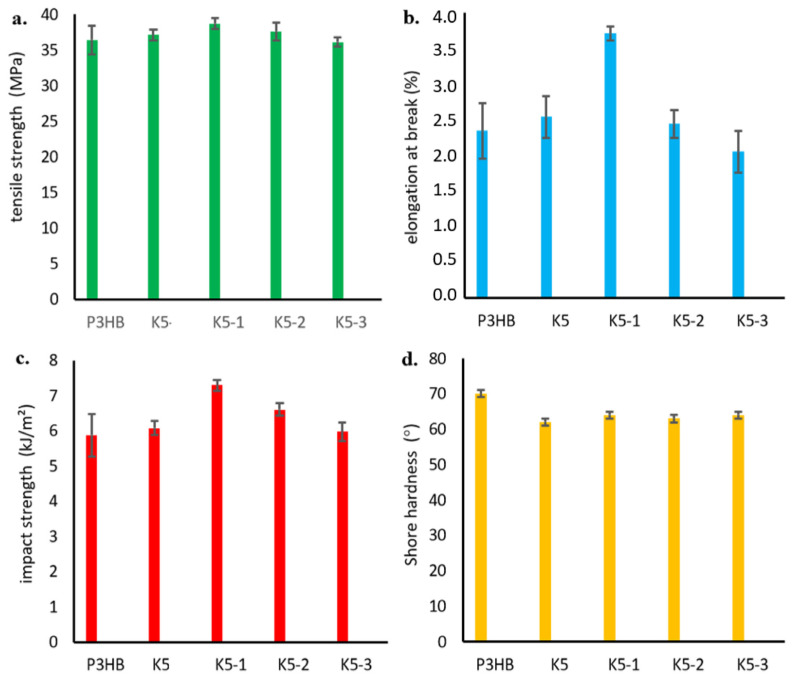
Dependency graph: (**a**) tensile strength, (**b**) relative elongation at break, (**c**) impact strength, and (**d**) hardness of the produced hybrid nanobiocomposites as a function of Cloisite^®^30B content.

**Figure 15 ijms-24-17405-f015:**
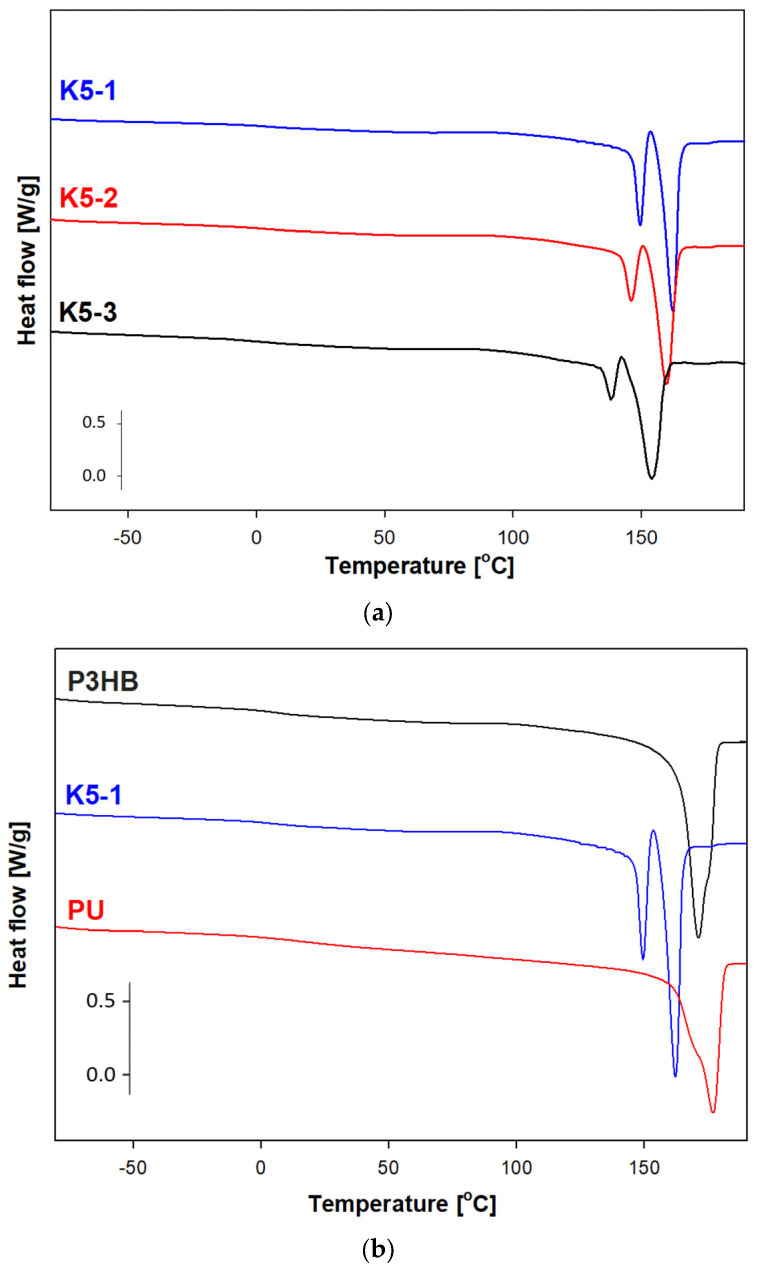
(**a**) Comparison of the heat flows of the nanobiocomposites versus temperature upon heating at a rate of 10 °C·min^−1^. (**b**) Comparison of the heat flow of poly(3-hydroxybutyrate) (P3HB), nanobiocomposite (K5-1) and polyurethane (PU) versus temperature.

**Table 1 ijms-24-17405-t001:** Composition of the polymer mixtures.

Content (m/m%)			Sample Designation
P3HB	PU	Cloisite^®^	
95	5	0	K5
90	10	0	K10
85	15	0	K15
80	20	0	K20
94	5	1	K5-1
93	5	2	K5-2
92	5	3	K5-3

**Table 2 ijms-24-17405-t002:** Interpretation of TG and DTG curves of the P3HB–PU polymer compositions and hybrid nanobiocomposites recorded at a heating rate of 5 °C/min under a nitrogen atmosphere.

Sample	T_on_ (°C)	T_50%_ (°C)	T_max1_ (°C)	Residue 600 °C (m/m%)
P3HB	221	246	278	1.01
K5	234	268	278	0.94
K10	266	277	278	0.75
K15	268	279	280	0.86
K20	268	279	279	0.64
K5-1	245	279	279	1.58
K5-2	245	279	278	5.64
K5-3	241	278	276	3.65

**Table 3 ijms-24-17405-t003:** Comparison of thermal parameters of P3HB–PU polymer blends upon heating their representative samples at 10 °C·min^−1^ after prior cooling at the same rate.

Sample	T_g_ (°C)	ΔC_p_ (J·g^−1^·°C^−1^)	T_m (onset)_ (°C)	T_m1_ (°C)	T_m2_ (°C)	ΔH_f_ (J·g^−1^)	T_c_ (°C)	ΔH_c_ (J·g^−1^)
P3HB	7.70	0.1620	159.73	165.75	-	91.93	90.50	88.79
K5	4.45	0.2257	149.92	153.30	165.30	92.98	88.00	79.06
K10	6.02	0.0680	145.03	150.95	163.50	88.14	85.60	68.98
K15	2.40	0.2294	140.81	144.55	158.10	94.55	80.80	67.04
K20	2.30	0.1875	135.53	140.12	153.20	92.67	76.40	66.00

**Table 4 ijms-24-17405-t004:** Comparison of the thermal parameters of nanobiocomposites upon heating their representative samples at 10 °C·min^−1^ after prior cooling at the same rate.

Sample	T_g_ (°C)	ΔC_p_ (J·g^−1^·°C^−1^)	T_m (onset)_ (°C)	T_m1_ (°C)	T_m2_ (°C)	ΔH_f_ (J·g^−1^)	T_c_ (°C)	ΔH_c_ (J·g^−1^)
K5-1	4.50	0.2059	146.70	149.50	162.20	90.75	86.70	68.34
K5-2	4.40	0.2328	142.00	146.00	159.80	89.78	82.87	68.97
K5-3	5.25	0.3752	134.60	138.20	153.80	85.70	77.65	63.58

## Data Availability

The data presented in this study are available on request from the corresponding author. The data are not publicly available due to privacy or ethical.

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
