# Peer review of "Polymer Biocompositions and Nanobiocomposites Based on P3HB with Polyurethane and Montmorillonite"

_ijms, 2023, doi:10.3390/ijms242417405_

Round 1
Reviewer 1 Report
Comments and Suggestions for Authors
The authors have prepared P3HB-PU polymer blends with different weight percentages (5, 10, 15, and 20 wt%) of PU and also prepared P3HB-PU (PU-5 wt%) blend nanocomposites using 1, 2, and 3 wt% of organically modified montmorillonite (Cloiste 30B) nanoclay as a filler. Upon blending P3HB with PU and making its nanocomposite using Cloisite 30B, improvements in thermal stability and mechanical properties have been reported in this work for a particular concentration of PU and Cloisite 30B in the blend and blend nanocomposite, respectively. However, this study lacks proper explanations of observed changes in many thermophysical properties.The following points need to be corrected or included:
1. This work did not report the miscibility of the P3HB-PU polymer blend, i.e., whether the polymer blend is miscible, immiscible, or partially miscible. Since the ultimate properties of the blend depend entirely on the miscibility of the blend, it is necessary to explain the observed changes in various thermophysical properties based on miscibility, which DSC measurements can do by measuring the glass transition temperatures of the component polymers and blends. For this reason, DSC measurement of pure Polyurethane (PU) should be taken and included in the manuscript. From the mechanical and thermal properties results, it looks like the blend is partially miscible; it needs to be confirmed.
2. In FTIR results, it is claimed that there is hydrogen bond interaction between the component polymer chains irrespective of blend compositions. On the other hand, to explain the decrease in mechanical properties like tensile strength, impact strength, and hardness for higher concentrations (>5 wt%) of PU, it is assumed that there is an increase in free volume in the blends. The above two results are contradictory.
3. The TGA and DTG graphs should be included in the manuscript. In another research article (https://doi.org/10.3390/nano13020225) from the same research group, it was reported that the decomposition onset temperature (Ton) of P3HB is 221°C, but in the present manuscript, it is 212 °C. However, the material, processing, and TGA measurement conditions are the same. How is it possible?
4. In DSC results, pure P3HB has only one melting peak. On the other hand, all the blends have two melting peaks. To explain these double melting peaks, authors have assumed it may be due to different crystalline forms. From the results, it is clear that the melting peaks at higher temperatures are due to P3HB, but the reason for the occurrence of melting peaks at lower temperatures is unclear. Either it is due to the melting of polyurethane, or the presence of polyurethane influences the crystal structure of P3HB. This needs to be confirmed by measuring and comparing the melting behavior of pure polyurethane. In Table 3, the authors have presented the enthalpy of melting or fusion (ΔHf) for each sample but did not mention whether these values are obtained from the first melting peaks, second melting peaks, or their combinations. Also, the changes in enthalpy of melting need proper explanations.
5. In Table 4, the melting onset temperature (Tm (onset)) of the K5-1 sample is higher than the first melting peak temperature (Tm1). It seems like a typo error, which needs to be corrected.
Comments on the Quality of English Language
None
Author Response
Thank you very much for your positive and friendly review.
1.This work did not report the miscibility of the P3HB-PU polymer blend, i.e., whether the polymer blend is miscible, immiscible, or partially miscible. Since the ultimate properties of the blend depend entirely on the miscibility of the blend, it is necessary to explain the observed changes in various thermophysical properties based on miscibility, which DSC measurements can do by measuring the glass transition temperatures of the component polymers and blends. For this reason, DSC measurement of pure Polyurethane (PU) should be taken and included in the manuscript. From the mechanical and thermal properties results, it looks like the blend is partially miscible; it needs to be confirmed.
Ad.1
Thank you for your comments. The Figure was added as follows:
Figure 15b. Comparison of heat flow of poly(3-hydroxybutyrate), P3HB, nanobiocomposite, K5-1, and polyurethane, PU versus temperature.
Figure 15b shows comparison of heat flow of poly(3-hydroxybutyrate), P3HB, nanobiocomposite, K5-1, and polyurethane, PU versus temperature. The PU heating scan presents the glass transition at Tg = 14.90°C and ΔCp = 0.3094 J·g-1·°C-1, and the melting region at Tm onset = 166.30°C and heat of fusion, ΔHf = 90.39 J·g. It can be observed that DSC heating scans presents a single glass transition and melting which indicates complete miscible of P3HB and PU in all nanocomposites.
- In FTIR results, it is claimed that there is hydrogen bond interaction between the component polymer chains irrespective of blend compositions. On the other hand, to explain the decrease in mechanical properties like tensile strength, impact strength, and hardness for higher concentrations (>5 wt%) of PU, it is assumed that there is an increase in free volume in the blends. The above two results are contradictory.
Ad. 2.
Thank you for your comment. It may seem that way, but these two explanation are not contradictory. The expression" an increase in free volume in the blends" refers to smaller interactions of P3HB chains. The additive of PU into P3HB results in interaction between PU and P3HB chains including occurrence of hydrogen bonds and simultaneously decreasing specific P3HB chains interactions (polyester/polyester interactions). Their presence causes the initial improvement of mechanical properties at a concentration of PU 5 m/m%. It means that hydrogen bonds are not completely responsible for improvement the mechanical properties of P3HB-PU compositions. It can be related with short chain of butanediol in PU structure and too stiff structure of polyurethane to proper elasticizing effect by PU.
- The TGA and DTG graphs should be included in the manuscript. In another research article (https://doi.org/10.3390/nano13020225) from the same research group, it was reported that the decomposition onset temperature (Ton) of P3HB is 221°C, but in the present manuscript, it is 212 °C. However, the material, processing, and TGA measurement conditions are the same. How is it possible?
Ad. 3. Thank you for your comment. The TGA and DTG curves were included in the manuscript – Figure 8.
You are right. It can wonder but it is possible, e. g., in DOI 10.1007/s10973-016-6039-9 decomposition onset temperature (Ton) of P3HB is 212°C (1% 215,4°C) whereas in doi: 10.37190/ABB-01987-2021-02 and doi: 10.37190/ABB-01782-2021-05 decomposition onset temperature (Ton) of P3HB is 236°C. This time, it was my mistake, a editorial error.
- In DSC results, pure P3HB has only one melting peak. On the other hand, all the blends have two melting peaks. To explain these double melting peaks, authors have assumed it may be due to different crystalline forms. From the results, it is clear that the melting peaks at higher temperatures are due to P3HB, but the reason for the occurrence of melting peaks at lower temperatures is unclear. Either it is due to the melting of polyurethane, or the presence of polyurethane influences the crystal structure of P3HB. This needs to be confirmed by measuring and comparing the melting behavior of pure polyurethane. In Table 3, the authors have presented the enthalpy of melting or fusion (ΔHf) for each sample but did not mention whether these values are obtained from the first melting peaks, second melting peaks, or their combinations. Also, the changes in enthalpy of melting need proper explanations.
Ad. 4. Thank you for your comment, but presence of two melting peaks is specific for poly(3-hydroxybutyrate) and it is dependent on the thermal history and no on PU. Please look at our previously paper where we are presenting quality and quantity (advanced) thermal analysis of double melting peak of poly(3-hydroxybutyrate). Below I show the our thermogram from following paper: Czerniecka-Kubicka A., Zarzyka I., Pyda M.: Advanced thermal analysis of poly(3-hydroxybutyrate), Przemysł Chemiczny 11 (2015) 45-49:
According to recommendation Wunderlich B. (Thermal Analysis of Polymeric Materials, Springer, Berlin, Heidelberg, New York 2005) for integration of double melting peak, it should be integration as following:
Hence, additional indication in the text of paper linking with integration of single or double melting peak are not required, because double melting peak is integrated for characteristic the whole melting region according to good integration practice.
Moreover, we added the DSC heating scan of PU to our paper as reference (please see Figure 15b in the manuscript). Influence of the PU and nanofiller addition on degree of crystallinity and other phase will be studied and discussed in our next paper.
- In Table 4, the melting onset temperature (Tm (onset)) of the K5-1 sample is higher than the first melting peak temperature (Tm1). It seems like a typo error, which needs to be corrected.
Ad. 5. Thank you for your comment. It was corrected.

Reviewer 2 Report
Comments and Suggestions for Authors
I support any work related to any green technology, like manuscript ijms-2711626 submitted by B. Krzykowska et al. (Polymer biocompositions and hybrid polymer nanobiocomposites...). This topic should be relevant and interesting for possible readers of the journal.
However, I have some serious concerns that should be addressed before publication. I suggest major revision before publication. Detailed comments are as follows.
Major concerns:
1) It is stated in the introduction that „Plastics are used in almost every field, from the automotive industry to medicine.” Which is certainly true and not questionable. I suggest specifying what are the specific application fields of the polimers investigated by the authors. Also, I advise specifying in the „Conclusions” why the results which are obtained improve their application possibilities.
2) There are some phrases which are unproper in scientific environment. E.g., what is the meaning of „properties are not satisfactory”, „improvement in thermal properties”, „weight average molecular mass”, „more formed crystals”, „nanocharge”, etc.
3) I am sorry, but I don’t know what I should see on TEM images in Figure 11.
4) In Figure 6 different samples are „designated K5, K10, K15 and K20, respectively”. However, it is not indicated.
5) Same problem in Figure 11.
6) Bottom of page 4: „... study the obtained nanostructured nanobiocomposites.” Is it not trivial that nanobiocomposites are nanostructured?
7) Unit marks should be indicated in figures whatever it is (%, W/g). What is the meaning of „[-]” in Figure 10?
8) I would prefer „m/m%” instead of „wt.%”.
Formal concerns:
There are large number of trivial typing mistakes. It seems that MS is prepared in a very superficial way. Authors should take much care of preparing the MS. If the MS is prepared in a superficial way, how can the reader trust the measurements?
1) The title is too long. I advise shortening it.
2) It seems that authors did not read the „Instruction for authors”. Abstract is much longer than allowed and the style does not fit.
3) Abbreviation of scientific phrases should be at their first appearance and uniform throughout the MS.
4) Check reference style in the text and in list of „References”.
5) Please check trivial typing and grammar mistakes.
My English is not native. However, it seems the readability of the text is not bad, grammatically mostly appears correct, however literally can be improved.
In conclusion, I suggest the MS to be accepted after major revision.
Author Response
Thank you very much for your positive and insightful review.
1) It is stated in the introduction that „Plastics are used in almost every field, from the automotive industry to medicine.” Which is certainly true and not questionable. I suggest specifying what are the specific application fields of the polimers investigated by the authors. Also, I advise specifying in the „Conclusions” why the results which are obtained improve their application possibilities.
Ad. 1. Thank you for your comment. The specific application fields of the investigated polymers was specified, in the Introduction.
The improved properties of obtained composites were emphasized and indicated their application possibilities, in the Conclusions.
2) There are some phrases which are unproper in scientific environment. E.g., what is the meaning of „properties are not satisfactory”, „improvement in thermal properties”, „weight average molecular mass”, „more formed crystals”, „nanocharge”, etc.
Ad. 2. Thank you for your comment. The confused phrases are corrected.
3) I am sorry, but I don’t know what I should see on TEM images in Figure 11.
Ad. 3. Thank you for your comment. The figures are provided with arrows identifying the exfoliated and intercalated structure.
4) In Figure 6 different samples are „designated K5, K10, K15 and K20, respectively”. However, it is not indicated.
Ad. 4. Thank you for your comment. It was corrected.
5) Same problem in Figure 11.
Ad. 5. Thank you for your comment. It was corrected.
6) Bottom of page 4: „... study the obtained nanostructured nanobiocomposites.” Is it not trivial that nanobiocomposites are nanostructured?
Ad. 6. Thank you for your comment. It was corrected.
7) Unit marks should be indicated in figures whatever it is (%, W/g). What is the meaning of „[-]” in Figure 10?
Ad. 7. Thank you for your comment. You are right. Units was indicated in Figure 10, presently in Figure 11.
8) I would prefer „m/m%” instead of „wt.%”.
Ad. 8. Thank you for your comment. It was changed.
Formal concerns
1) The title is too long. I advise shortening it.
Ad. Thank you for your comment. 1. It was shortened.
2) It seems that authors did not read the „Instruction for authors”. Abstract is much longer than allowed and the style does not fit.
Ad. 2. Thank you for your comment. Abstract was corrected.
3) Abbreviation of scientific phrases should be at their first appearance and uniform throughout the MS.
Ad. 3. Thank you for your comment. It was checked.
4) Check reference style in the text and in list of „References”.
Ad. 4. Thank you for your comment. It was checked.
5) Please check trivial typing and grammar mistakes.
Ad. 5. Thank you for your comment. It was checked.

Round 2
Reviewer 2 Report
Comments and Suggestions for Authors
Manuscript “ijms-2711626 -peer-review-v2” submitted by B. Krzykowska et al. (Polymer biocompositions and nanobiocomposites based on ...) is the revised version of the MS of the same article number submitted by the same authors.
Regarding the significance of the topic and its suitability for the scope of the journal I can confirm my earlier statement.
It was really disturbing that the pdf conversion was not correct, and I found so many formal mistakes, also in this second submission by reading it. However, authors provided the source word file in which the readability of the MS was much better. Now I am using this source file for reviewing. I just note that the source file was presented under the name of cover letter, which was actually, missing.
The scientific sound of the MS improved a lot in the revised version and most of the earlier formal mistakes were corrected. I have only a few formal notes.
1. I still believe that units should be provided, either by marks on the axes of figures or by unit bars inside the graphs. It is important to indicate how big is the transmittance (1, 0.1, 0.001, 001” etc.), or the given physical/chemical parameter (e.g., Heat flow (W/g)). By the way, transmittance is double „t” (Figs. 3, 10).
2. Same problem wherever units are missing on axes. The marks also should be provided (e.g. og Figs. 7 and 14).
3. I still don’t understand what is the meaning of “(-)” (Figs. 7d, 14d,).
4. „hardness Shore” (Fig 7d) or „Shore hardness” (Fig 14d)?
5. I would reduce the thicknesses of the arrows on Fig 12 (too dominating arrows are here).
6. There are some formal mistakes in the list of references. E.g., „Chen, G.Q.”, “Ibrahim, NA.”.
My suggestion is publishing after minor changes.
Author Response
Thank you for your comments.
- I still believe that units should be provided, either by marks on the axes of figures or by unit bars inside the graphs. It is important to indicate how big is the transmittance (1, 0.1, 0.001, 001” etc.), or the given physical/chemical parameter (e.g., Heat flow (W/g)). By the way, transmittance is double „t” (Figs. 3, 10).
Ad. 1. They were corrected.
- Same problem wherever units are missing on axes. The marks also should be provided (e.g. og Figs. 7 and 14).
Ad. 2. It was corrected.
- I still don’t understand what is the meaning of “(-)” (Figs. 7d, 14d,).
Ad. 3. It was corrected.
- „hardness Shore” (Fig 7d) or „Shore hardness” (Fig 14d)?
Ad. 4. It was corrected. Shore hardness is correct.
- I would reduce the thicknesses of the arrows on Fig 12 (too dominating arrows are here).
Ad. 5. It was corrected.
- There are some formal mistakes in the list of references. E.g., „Chen, G.Q.”, “Dias, Y. J.”, „Ibrahim, NA.”.
Ad. 6. It was corrected.
